# A Combination of Aspirin and Clopidogrel Predict More Favorable Dynamics of Platelet Reactivity versus Clopidogrel Alone in the Acute Phase of Minor Stroke

**DOI:** 10.3390/healthcare9060628

**Published:** 2021-05-25

**Authors:** Adam Wiśniewski, Joanna Sikora, Aleksandra Karczmarska-Wódzka, Przemysław Sobczak

**Affiliations:** 1Department of Neurology, Collegium Medicum in Bydgoszcz, Nicolaus Copernicus University in Toruń, 85-094 Bydgoszcz, Poland; 2Experimental Biotechnology Research and Teaching Team, Department of Transplantology and General Surgery, Collegium Medicum in Bydgoszcz, Nicolaus Copernicus University in Toruń, 85-094 Bydgoszcz, Poland; joanna.sikora@cm.umk.pl (J.S.); akar@cm.umk.pl (A.K.-W.); przemyslawsobczak02@gmail.com (P.S.)

**Keywords:** stroke, cerebrovascular diseases, prevention, dual antiplatelet therapy, clopidogrel, aspirin, platelet reactivity

## Abstract

Background: The combined use of clopidogrel and aspirin is recommended for the short-term (21 days) therapy of minor stroke or transient ischemic attack. Previous studies have demonstrated its efficacy and superiority over treatment with a single antiplatelet agent. However, there is insufficient support for the advantages of such therapy based on platelet function testing. We aimed to compare the effect of the concomitant use of clopidogrel and aspirin versus clopidogrel alone on the dynamics of platelet reactivity over time to determine the appropriate antiplatelet treatment strategy for minor strokes. Methods: We enrolled 74 ischemic stroke subjects, including 38 minor strokes. Platelet reactivity was assessed by impedance aggregometry (Multiplate Analyzer) 48 and 96 h after a first 75 mg dose of clopidogrel, using the acetylsalicylic acid platelet inhibition (ASPI) test and the adenosine diphosphate (ADP) test. Dual antiplatelet therapy was strictly reserved only to minor strokes, as the other strokes received clopidogrel alone in the secondary prevention. The dynamics of platelet reactivity refer to the difference between two assessments, and a decrease in values over time was considered favorable. Results: The incidence of clopidogrel non-responsiveness was 64.8%, and this was similar in the group of minor strokes and the group of more disabling strokes. We indicated diabetes mellitus as an independent predictor of high on-clopidogrel platelet reactivity (Odds ratio OR 5.69 95% Confidence Interval CI 1.13–41.26, *p* = 0.0386). Among minor strokes treated with dual antiplatelet therapy, in relation to clopidogrel, we reported a trend toward more favorable dynamics of platelet reactivity over time compared to the group using clopidogrel alone (*p* = 0.0652 vs. *p* = 0.3384, respectively). We identified five predictors (sex, female; small-vessel disease; no diabetes; no hyperlipidemia; and no alcohol abuse) related to a significant decrease in platelet reactivity over time with respect to clopidogrel. No significant dynamics of platelet reactivity when using aspirin were found. Conclusions: Our findings, based on the favorable dynamics of platelet reactivity over time in relation to clopidogrel, confirm the usefulness of dual antiplatelet therapy in minor strokes and support the continuation of the secondary prevention with clopidogrel alone rather than aspirin, particularly among identified beneficiaries of such a strategy.

## 1. Introduction

Dual antiplatelet therapy has become the standardized strategy in myocardial infarction. Recently, it is also increasingly being used in the secondary prevention of minor ischemic stroke or transient ischemic attack. The updated guidelines regarding acute stroke management recommend the combined use of clopidogrel and aspirin for a period of 21 days, then the continuation of treatment with one of the antiplatelet agents alone [1]. Several large clinical trials have demonstrated that dual antiplatelet therapy is superior compared to single drug treatment and have emphasized the safety and efficacy of such a combination [2,3,4]. However, the simultaneous use of aspirin and clopidogrel is strictly limited in terms of time and reserved only for individuals with non-disabling strokes.

The effective inhibition of platelet function is essential for the proper prevention of cerebrovascular accidents. Different platelet function assays enable a fast, reliable, and comparable assessment of the platelet inhibition rates by the antiaggregant [5]. Nonetheless, the phenomenon of non-responsiveness or high on-treatment platelet reactivity is common and significantly reduces the efficacy of antiplatelet therapy, contributing a to worse prognosis or higher incidence of recurrent ischemic events [6,7,8,9]. The prevalence is estimated at 5–65% and is comparable for both clopidogrel and aspirin [10,11]. Similar mechanisms underlying the altered biological response are responsible for the co-resistance of up to 50% [12].

Considering the different platelet inhibition mechanisms, target points, and metabolic pathways, when a combination of treatments is used, a synergistic and complementary effect should be expected. Previous studies investigating myocardial infarction have demonstrated that the combined use of clopidogrel and aspirin is superior in terms of platelet inhibition compared to the effect of a single agent [13]. Similar findings were also reported for ischemic stroke [14,15,16]. This translates into a more effective reduction in the risk of recurrent cerebrovascular incidents [17]. However, this assessment of platelet reactivity was not only performed exclusively among minor strokes. Furthermore, the authors compared platelet function before and after treatment initiation, often after an acute phase of stroke, and did not focus on factors that could influence the degree of inhibition. These deficiencies and disadvantages prompted us to conduct a study monitoring the effectiveness of dual antiplatelet therapy in the acute phase of stroke based on the assessment of the dynamics of platelet function only in a minor stroke group.

The aim of the current study was to estimate the effect of dual antiplatelet therapy on the dynamics of platelet function compared to the use of clopidogrel alone in the acute phase of minor stroke, as well as to determine the predictors of favorable and effective changes in platelet reactivity, reflecting proper platelet inhibition.

## 2. Materials and Methods

### 2.1. Study Design and Participants

This prospective and observational study was conducted from 1 October 2019 to 31 December 2020 in a Stroke Center in the Department of Neurology at the University Hospital No. 1 in Bydgoszcz, Poland. We enrolled 74 ischemic stroke subjects. The severity of stroke was assessed by the National Institutes of Health Stroke Scale (NIHSS). Participants with 0–3 points (*n* = 38) on admission were included to the minor stroke group and received 300 mg of clopidogrel and 75 mg of aspirin on the first day and from the second day 75 mg of aspirin and 75 mg of clopidogrel. The others (*n* = 36) with NIHSS >3 points, were treated with 150 mg of aspirin on the first day and from the second day with 75 mg of clopidogrel alone. A comparison of the baseline characteristics of the stroke subjects in both groups is presented in Table 1.

We used the following exclusion criteria: specific stroke therapy (intravenous thrombolysis and/or endovascular treatment), documented cardioembolic background of stroke (e.g., atrial fibrillation), subjects with severe speech disorders or impaired consciousness (inability to sign the informed consent), advanced oncological disease, urinary or gastrointestinal bleeding in the previous year, previous stroke history in the last 2 years, thrombocytopenia below 100 thousands/µL and taking antiplatelet drugs before the current stroke incident.

### 2.2. Platelet Reactivity Research

Impedance aggregometry (the Multiplate–Dynabyte multichannel platelet function analyzer, Roche Diagnostics, France) was performed to evaluate platelet function in the Laboratory of Experimental Biotechnology at Collegium Medicum in Bydgoszcz. Multiplate Analyzer is mentioned as one of the four referenced and recommended point-of-care methods for assessing platelet function, that are considered equivalent [18]. The choice of a platelet function assay depends on the availability and experience of the center. We applied a standardized and validated assay and adopted cutoff values in accordance with the current knowledge and guidelines in this field. We collected two blood samples from the veins of the forearm. The first assessment was 48 h (+/−4 h) after the first 75 mg dose of clopidogrel. The second measurement was assessed 96 h (+/−4 h) after the first 75 mg dose of clopidogrel. An acetylsalicylic acid platelet inhibition (ASPI) test and adenosine diphosphate (ADP) test were applied in this study, with arachidonic acid and ADP as the agonists of the platelets, respectively. The addition of an activator to the solution mobilized the platelets to move towards two electrodes, which was received by device system as a change in resistance (impedance). Then, electrical signals were transformed automatically into graphical figures, where resistance differences were converted to area under the curve units (AUC). The average for two electrode pairs was the final result of the platelet function measurement. As all the stroke subjects underwent an ADP test during the clopidogrel treatment, we set values over 46 AUC as the cut-off point to confirm the high on-treatment platelet reactivity, reflecting non-responsiveness to clopidogrel. Other authors have also adopted a similar value in their studies [19,20]. The processing steps of the platelet aggregation measurements were conducted according to the guidelines reported by other investigators [21,22]. The differences in values between two assessments were considered as the dynamics of platelet reactivity over time, and a decrease in values was reported as a favorable change.

### 2.3. Vascular Risk Factors

We adopted the following criteria for vascular risk factors: hypertension (documented and recognized before enrollment or at least two measurements over 140/90 mmHg during current hospitalization), diabetes (documented and recognized before enrollment or at least two measurements over 200 mg% for serum glucose during current hospitalization), hyperlipidemia (documented and recognized before enrollment or a level of cholesterol over 180 mg/dL during current hospitalization), obesity (Body Mass Index over 30 kg/m^2^), smoking (current smokers), alcohol abuse (at least two beers or 100 mL of vodka over 15 days per month in the last 6 months), large-vessel disease (at least 50% stenosis of artery responsible for stroke symptoms), and small-vessel disease (typical morphological lesions in the deep white matter of the brain in neuroimaging). Laboratory parameters were assessed in the Department of Laboratory Diagnostics in the University Hospital No. 1 in Bydgoszcz. The blood samples were collected from the veins of the forearm on the first day of the stroke at 6 a.m.

### 2.4. Ethical Statement

All the participants read the study protocol and signed the informed consent before enrollment. The study protocol was approved by the Bioethics Committee of the Nicolaus Copernicus University in Torun at Collegium Medicum of Ludwik Rydygier in Bydgoszcz (KB number 704/2019) on 24.09.2019. This research was conducted in accordance with the Declaration of Helsinki.

### 2.5. Statistical Evaluation Methods

The statistical analysis was carried out with the STATISTICA device, version 13.1 (Dell company, Round Rock, TX, USA). The collected data were expressed with non-parametric characteristics as medians and ranges. The Mann–Whitney U test was used for the estimation of the dynamics of platelet reactivity over time, Fisher’s exact test was used to compare the baseline parameters of stroke subjects and McNemar’s chi-squared test with continuity correction. was used to value differences among parameters in timepoints. Univariate and multivariate logistic regression models were used for the evaluation of predictors related to clopidogrel non-responsiveness. The level of *p* < 0.05 was considered statistically significant.

## 3. Results

We achieved a clopidogrel non-responsiveness of 64.8% among all stroke subjects. The incidence did not differ between minor stroke subjects and others (Table 1).

The only significant differences between the two subgroups were the severity and disability of stroke on admission. Using univariate logistic regression, we revealed that diabetes, obesity, and levels of platelets and hemoglobin A1c were associated with the risk of high on-clopidogrel platelet reactivity (Table 2).

Using multivariate logistic regression, we reported diabetes as the only independent predictor of clopidogrel non-responsiveness in a model adjusted for significant variables in univariate analysis (Odds ratio OR 5.69 95% Confidence Interval CI 1.13–41.26, *p* = 0.0386).

In the group on clopidogrel alone, we showed an insignificant decrease in platelet reactivity over time (from a median value of 53 to 47.5 AUC, *p* = 0.3384) (Figure 1A).

The etiology of stroke, sex, and cardiovascular risk factors did not significantly influence the dynamics of platelet reactivity during therapy with clopidogrel alone (Table 3).

Similarly, among patients with minor strokes on dual antiplatelet therapy, we reported a slight decrease in platelet reactivity over time in relation to aspirin (from median value 19.5 to 18 AUC, *p* = 0.6180). By contrast, in relation to clopidogrel, we showed a trend toward more favorable dynamics of platelet reactivity over time compared to the group treated with clopidogrel alone (from a median value of 50.5 to 44.5 AUC, *p* = 0.0652) (Figure 1B). In addition, we identified five favorable factors (sex, female; small-vessel disease; no diabetes; no hyperlipidemia; and no alcohol abuse), which contributed to achieving a significant and beneficial decrease in platelet reactivity over time with respect to clopidogrel (Figure 2A,C,E,G,I). Conversely, in male subjects with diabetes and alcohol abuse we observed unfavorable changes in the form of an increase in platelet reactivity over time (Figure 2D,H,J).

However, no similar impact on the dynamics of platelet reactivity were found in relation to aspirin (Table 4).

Nevertheless, factors related to significant changes regarding clopidogrel were also largely associated with a favorable decrease in platelet reactivity in relation to aspirin, whereas their opposites were associated with unfavorable increases in values over time (excluding diabetes).

## 4. Discussion

To the best of our knowledge, we are the first to assess the dynamics of platelet reactivity in the acute phase of minor stroke and highlight that dual antiplatelet therapy promotes more favorable changes in platelet reactivity over time with respect to clopidogrel than aspirin. Furthermore, the use of dual antiplatelet therapy causes more beneficial decreases in values of platelet activity compared to clopidogrel alone. Moreover, we emphasized the predictors which enabled the dynamics of platelet reactivity to reach significance in relation to clopidogrel. Our findings allow us to hypothesize that, for the specific subgroups of minor strokes, dual antiplatelet therapy remains essential and the continuation of the secondary prevention with clopidogrel alone after 21 days is more reasonable and justified than with aspirin. We suggest that pathophysiological mechanisms responsible for aspirin resistance may be less susceptible to external factors as compared to the mechanisms determining clopidogrel resistance. Therefore, it might have been a greater tendency to “break” the resistance among high on-clopidogrel platelet reactivity in contrast to high on-aspirin platelet reactivity subjects. However, our theory must be verified in further research.

In contrast to many reports demonstrating the efficacy, safety, and beneficial properties of dual antiplatelet therapy with the combined use of clopidogrel and aspirin compared to the use of a single antiplatelet agent [2,3,4,17,23,24], only a few have raised this topic in terms of platelet reactivity. Kang et al. [14] reported that the values of platelet reactivity were significantly lower in the group treated with dual antiplatelet therapy compared to the groups treated with aspirin or clopidogrel alone. However, the authors did not focus only on minor strokes; used another device for platelet function testing (turbidimetric aggregometry); and did not analyze the dynamics, assessing only a single time-point measurement. Yi et al. [15] showed that the inhibition of platelets during dual antiplatelet therapy is superior to that obtained using clopidogrel alone due to the synergistic effect on both arachidonic acid- and ADP-induced platelet aggregation. However, in contrast to our findings, no trend has been detected showing a more effective platelet inhibition based on ADP-induced platelet aggregation during dual antiplatelet therapy (*p* = 0.871). Similar to our study, they performed a double assessment of platelet reactivity, but the first measurement was before the treatment and the second was 7–10 days later. Therefore, the dynamics of platelet function on dual antiplatelet therapy were not captured. Furthermore, the authors analyzed all stroke subjects with NIHSS total scores below 15 rather than focusing on minor strokes and performed platelet function based on light transmission aggregometry. In another study, Yi et al. [16] compared the platelet reactivity between stroke subjects on dual antiplatelet therapy versus aspirin alone. They reported significantly lower values of platelet aggregation one month after dual antiplatelet therapy compared to when using aspirin alone (*p* < 0.001). In contrast to our study, the authors did not analyze the dynamics of platelet reactivity in the acute phase of stroke and included stroke subjects with NIHSS total scores below 13 in dual antiplatelet therapy. Moreover, they included only subjects with large artery atherosclerosis in their etiology of stroke.

The percentage of clopidogrel non-responders in our study was relatively high and was located in the upper limit of the values reported by other authors. However, Kinsella et al. [25] showed a 92 percent incidence of clopidogrel non-responsiveness among stroke subjects. We revealed that the occurrence of high on-clopidogrel platelet reactivity achieved similar values in minor strokes compared to more severe strokes. We highlighted that diabetes mellitus, obesity, higher count of platelets, and hemoglobin A1c values are associated with the risk of clopidogrel non-responsiveness. Our findings are consistent with results obtained by other authors [19,26,27,28]. However, diabetes mellitus remained the only independent predictor in multivariate analysis and the only one in the prognostic model without laboratory findings. A detailed explanation of the mechanisms underlying the influence of diabetes mellitus on the occurrence of clopidogrel resistance has been demonstrated in many outstanding studies [29,30,31].

It is worth mentioning the recent reports on the superior effect of the combined use of ticagrelor and aspirin compared to clopidogrel and aspirin in minor strokes and transient ischemic attacks, which also showed positive results in terms of platelet reactivity [32,33,34]. The authors emphasized not only the efficacy and safety of such a combination, but additionally proved the significantly more effective inhibition of platelets, both in the acute phase of a cerebrovascular event as well as at 3 months. Moreover, they indicated the subgroups (e.g., large artery atherosclerosis etiology of stroke) that could especially benefit from the combined therapy with ticagrelor and aspirin [33]. However, further studies are needed in this topic, as the concomitant use of ticagrelor and aspirin is still not officially recommended for the treatment of minor strokes.

We are fully aware of some limitations regarding our study. Our findings are based on a small sample size that should be confirmed using a larger population. There is a lack of a widely accepted and standardized measurement of platelet function. Our research does not include subjects with symptoms of severe aphasia or impaired consciousness due to the inability to obtain informed consent from these subjects. Further analysis of platelet function is necessary also among stroke subjects who have undergone reperfusion therapy.

## 5. Conclusions

In summary, the novel findings shown in the current study support the combined use of clopidogrel and aspirin rather than a single antiplatelet agent in minor strokes with respect to the more favorable dynamics of platelet reactivity obtained over time. Moreover, emphasizing the significantly greater inhibition of platelets in relation to clopidogrel in such a combination, we provide support for choosing clopidogrel as a continuation of the secondary prevention after three weeks of dual antiplatelet therapy in minor strokes. Furthermore, the proposed antiplatelet therapy strategy should be considered primarily in the indicated specific subgroups of minor strokes (females with small-vessel disease with no diabetes, hyperlipidemia, or alcohol abuse) that could especially benefit from optimized, platelet function-guided therapy.

## Figures and Tables

**Figure 1 healthcare-09-00628-f001:**
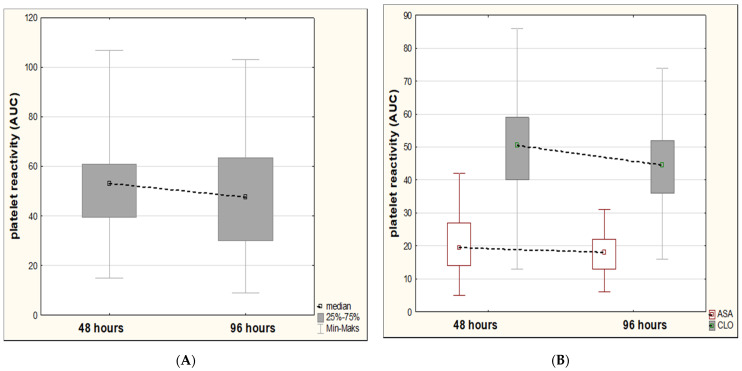
The overall dynamics of platelet reactivity (in area under the curve units, AUC) over time. (**A**) The group on clopidogrel alone. The insignificant decrease in platelet reactivity between two assessments at 48 and 96 h. (**B**) The group on dual antiplatelet therapy. The insignificant decrease in platelet reactivity on aspirin (ASA) and a trend towards a more favorable decrease in platelet reactivity on clopidogrel (CLO).

**Figure 2 healthcare-09-00628-f002:**
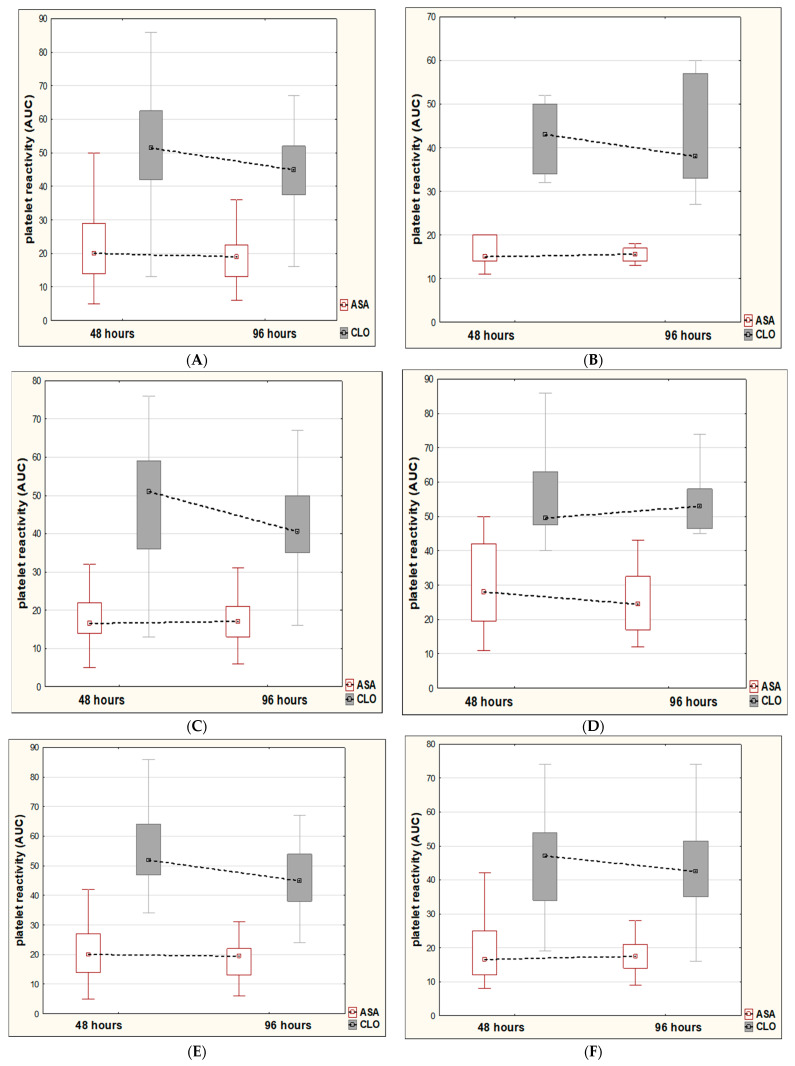
Predictors of significant and favorable dynamics of platelet reactivity (in area under the curve units, AUC) over time in the dual antiplatelet therapy group. The significant and favorable decrease in platelet reactivity between two assessments in relation to clopidogrel (CLO) were reported in subjects with small-vessel disease (**A**), an absence of diabetes mellitus (**C**), an absence of hyperlipidemia (**E**), female sex (**G**), and an absence of alcohol abuse (**I**). The dynamics of platelet reactivity in relation to aspirin (ASA) showed a favorable decrease in the majority of the above cases. The insignificant and largely unfavorable increase in platelet reactivity between two assessments in relation to clopidogrel (CLO) were reported in subjects with large-vessel disease (**B**), diabetes mellitus (**D**), hyperlipidemia (**F**), male sex (**H**), and alcohol abuse (**J**). The dynamics of platelet reactivity in relation to aspirin (ASA) showed an unfavorable increase in the majority of the above cases.

**Table 1 healthcare-09-00628-t001:** Comparison of the baseline characteristics of the groups with a single and dual antiplatelet therapy.

Parameter	CLO (*n* = 36)	CLO + ASA (*n* = 38)	*p*-Value
Age:	73.5 (18–91)	64 (34–89)	0.1690
Sex:			
Male	19 (52.7%)	17 (44.7%)	0.3233
Female	17 (47.3%)	21 (55.3%)	0.4256
Stroke etiology:			
Large-vessel disease	12 (33.3%)	6 (15.8%)	0.0681
Small-vessel disease	24 (66.7%)	32 (84.2%)	0.2356
Diabetes	13 (36.1%)	8 (21.0%)	0.1193
Hyperlipidemia	11 (30.6%)	16 (42.1%)	0.2150
Hypertension	28 (77.8%)	25 (65.8%)	0.1882
Obesity	8 (22.2%)	14 (36.8%)	0.1311
Smoking	12 (33.3%)	10 (26.3%)	0.3425
Alcohol abuse	5 (13.9%)	2 (5.3%)	0.1930
Clopidogrel non-responsiveness	23 (63.9%)	25 (65.8%)	0.5286
Platelet count	254 (142–618)	274.5 (115–414)	0.7468
HBA1C	6.05 (5.0–10.2)	5.7 (4.9–13.1)	0.5268
D-dimer	335 (165–2560)	357.5 (170–2455)	0.5892
CRP	2.78 (0.45–88.2)	2.26 (0.32–73.05)	0.4578
Fibrinogen	311 (202–657)	329 (198–658)	0.5824
NIHSS on admission	5 (1–16)	2 (1–3)	<0.0001
mRS on admission	3 (1–5)	1 (1–3)	<0.0001

*p*-values are measured accordingly with Mann–Whitney U-test or Fisher’s exact test. NIHSS—the National. Institute of Health Stroke Scale; mRS- modified Rankin scale; CRP—C-reactive protein; HBA1c—hemoglobin A1c; ASA—aspirin; CLO—clopidogrel.

**Table 2 healthcare-09-00628-t002:** Univariate and multivariate logistic regression analysis of predictors related to clopidogrel non-responsiveness.

Parameter	Univariate Analysis		Multivariate Analysis	
	OR (95% CI)	*p*	Adjusted OR (95% CI)	*p*
Sex (female)	1.91 (0.73, 5.02)	0.1899		
Age	0.98 (0.94, 1.02)	0.3740		
Diabetes	7.86 (1.66, 37.2)	0.0093 *	5.69 (1.13, 41.26)	0.0386 *
Large-vessel disease	0.44 (0.18, 1.29)	0.1338		
Smoking	1.50 (0.36, 6.23)	0.5767		
Hyperlipidemia	0.53 (0.20, 1.42)	0.2061		
Hypertension	1.20 (0.42, 3.41)	0.7372	2.35 (0.92, 7,68)	0.0964
Obesity	3.60 (1.07, 12.11)	0.0382 *		
Alcohol abuse	0.70 (0.14, 3.38)	0.6542	1.01 (0.99, 1.02)	0.1956
Platelet count	1.01 (1.00 1.02)	0.0464 *		
CRP	1.01 (0.96, 1.05)	0.8293	2.65 (0.92, 8.92)	0.1568
HBA1c	3.75 (1.19, 11.82)	0.0242 *		
Fibrinogen	1.01 (0.99, 1.02)	0.0955		
D-Dimer	1.00 (0.99, 1.00)	0.5698		

*—significant dependencies, OR—odds ratio, CI—confidence interval, CRP—C-reactive protein, HBA1c—hemoglobin A1c.

**Table 3 healthcare-09-00628-t003:** The dynamics of platelet reactivity in the group with a single antiplatelet agent (clopidogrel).

Parameter	Median 48 h	Median 96 h	*p*-Value
Overall	53 (15–107)	47.5 (9–103)	0.3384
Sex:			
Male	52 (15–107)	46 (9–103)	0.5019
Female	56 (20–86)	49 (15–86)	0.4590
Stroke etiology:			
Large-vessel disease	47 (20–86)	51 (19–86)	0.9081
Small-vessel disease	56 (15–107)	46 (9–103)	0.2048
Diabetes:			
Yes	58 (43–107)	66 (9–103)	0.6081
No	44 (15–79)	38 (15–86)	0.4290
Hyperlipidemia:			
Yes	52 (20–86)	46 (9–83)	0.4502
No	56 (15–107)	48 (15–103)	0.4492
Hypertension:			
Yes	51 (15–107)	45.5 (9–103)	0.2868
No	57.5 (26–79)	51.5 (17–86)	0.8336
Obesity:			
Yes	59.5 (20–86)	61 (47–83)	0.5286
No	46 (15–107)	41 (9–103)	0.3215
Smoking:			
Yes	57 (20–86)	50 (9–83)	0.5067
No	49 (15–107)	46.5 (15–103)	0.5567
Alcohol abuse:			
Yes	58 (36–71)	66 (29–77)	0.2154
No	52 (15–107)	46 (9–1030	0.6761

**Table 4 healthcare-09-00628-t004:** Comparison of the dynamics of platelet reactivity between aspirin and clopidogrel in the minor stoke group with dual antiplatelet therapy.

Parameter	Median 48 h	ASA Median 96 h	*p*-Value	Median 48 h	CLO Median 96 h	*p*-Value
Overall	19.5 (5–50)	18 (6–43)	0.6180	50.5 (13–86)	44.5 (16–74)	0.0652
Sex:						
Male	18 (5–50)	19.5 (9–36)	0.6647	44.5 (19–86)	45 (20–57)	0.7630
Female	20 (7–42)	16.5 (6–43)	0.3072	53 (13–76)	44 (16–74)	0.0399 *
Stroke etiology:						
Large-vessel disease	15 (11–20)	15.5 (13–18)	0.9989	43 (32–52)	38 (27–60)	0.9899
Small-vessel disease	20 (5–50)	19 (6–43)	0.5912	51.5 (13–86)	45 (16–74)	0.0333 *
Diabetes:						
Yes	28 (11–50)	24.5 (14–43)	0.7131	49.5 (40–86)	53 (45–74)	0.8336
No	16.5 (5–41)	17 (6–31)	0.7788	51 (13–86)	40.5 (16–67)	0.0459 *
Hyperlipidemia:						
Yes	16.5 (8–42)	17.5 (9–43)	0.9850	47 (19–47)	42.5 (16–74)	0.5591
No	20 (5–50)	19.5 (6–36)	0.4596	52 (13–86)	45 (24–67)	0.0423 *
Hypertension:						
Yes	20 (5–50)	18 (6–43)	0.4849	52 (19–86)	45 (20–74)	0.0625
No	15 (7–28)	19 (10–31)	0.8175	50 (13–70)	44 (16–67)	0.6081
Obesity:						
Yes	16 (5–50)	18.5 (6–43)	0.7652	55.5 (23–86)	48 (20–74)	0.1904
No	20 (7–42)	17.5 (9–29)	0.2836	50 (13–76)	44 (16–57)	0.1639
Smoking:						
Yes	20 (8–38)	17.5 (6–28)	0.4727	52 (32–73)	40.5 (33–57)	0.1405
No	17.5 (5–50)	18 (9–43)	0.8570	50 (13–86)	45.5 (16–74)	0.2253
Alcohol abuse:						
Yes	13.5 (11–16)	15.5 (13–18)	0.6985	42 (34–50)	50.5 (41–60)	0.6985
No	20 (5–50)	18 (6–43)	0.5431	51 (13–86)	44.5 (16–74)	0.0382 *

*—significant dependencies, ASA—aspirin, CLO—clopidogrel.

## Data Availability

The data that support the findings of this study are available from the corresponding author upon reasonable request.

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
