# Peer review of "A Combination of Aspirin and Clopidogrel Predict More Favorable Dynamics of Platelet Reactivity versus Clopidogrel Alone in the Acute Phase of Minor Stroke"

_healthcare, 2021, doi:10.3390/healthcare9060628_

Round 1

Reviewer 1 Report

Great paper. 

Would like to read a bit more about why no significant dynamics of platelet reactivity on aspirin was found.  Include more about the theory/reasoning for that perhaps? 

Also more literature on why diabetes is a predictor of high on-clopidogrel platelet reactivity.

Reviewer 2 Report

This study tried to compare the effect on platelet reactivity of dual antiplatelet therapy (Aspirin + clopidogrel) vs clopidogrel alone in patient with minor stroke. The study is interesting but lack of precision in methods and statistical analysis. I have some concerns:

  • Please describe in abstract that platelet reactivity was tested with multiplate via ASPI and ADP test and timing of the assessment.
  • Which was the dose of aspirin after the first day? Always 150 mg?
  • Please add “kg/m2” for BMI cut-off of obesity in methods.
  • Please describe how the assessment of PR with Multiplate would have influenced the results in comparison to other systems to evaluate platelet aggregation (e.g. light transmission aggregometry, Verifynow system).
  • Were patients already in chronic treatment with antiplatelet excluded?
  • How did the author choose the variables to be included in multivariate models? Why did authors add also CRP in model 2 of multivariate analysis if p in univariate was >0.05? Usually only variables statistically significant at univariate analysis have to added to the multivariate model. Please repeat analysis choosing only variables statistically significant at univariate.
  • Please convert table 2 and 3 in just one table with univariate and multivariate analysis, underlying better which variables were statistically significant in univariate and thus deserve to be included in multivariate.
  • Mann Whithney U might have been used to compare variables with non-continuous distribution, but which test was used to value differences among parameters in different timepoint (e.g.  McNemar's Chi-squared test with continuity correction)?

Reviewer 3 Report

Do you have the data before using antiplatelet agents 

Round 2

Reviewer 2 Report

Authors replied to all my questions. I have no other suggestions. 

This manuscript is a resubmission of an earlier submission. The following is a list of the peer review reports and author responses from that submission.